# Cause of Death, Mortality and Occult Blood in Colorectal Cancer Screening

**DOI:** 10.3390/cancers14010246

**Published:** 2022-01-04

**Authors:** Lasse Kaalby, Issam Al-Najami, Ulrik Deding, Gabriele Berg-Beckhoff, Robert J. C. Steele, Morten Kobaek-Larsen, Aasma Shaukat, Morten Rasmussen, Gunnar Baatrup

**Affiliations:** 1Department of Surgery, Odense University Hospital, 5000 Odense, Denmark; Issam.Al-Najami@rsyd.dk (I.A.-N.); Ulrik.Deding@rsyd.dk (U.D.); Morten.Kobaek.Larsen@rsyd.dk (M.K.-L.); Gunnar.baatrup@rsyd.dk (G.B.); 2Department of Clinical Research, University of Southern Denmark, 5000 Odense, Denmark; 3Unit for Health Promotion Research, Department of Public Health, University of Southern Denmark, 6700 Esbjerg, Denmark; gbergbeckhoff@health.sdu.dk; 4Unit for Health Research, Hospital South West Jutland, 6700 Esbjerg, Denmark; 5Centre for Research into Cancer Prevention and Screening, University of Dundee School of Medicine, Dundee DD1 9SY, UK; r.j.c.steele@dundee.ac.uk; 6GI Section, Minneapolis VA Medical Center and University of Minnesota, Minneapolis, MN 55417, USA; aasma.shaukat@va.gov; 7Division of Gastroenterology NYU Langone, New York, NY 10016, USA; 8Digestive Disease Center, Bispebjerg University Hospital, 2400 Copenhagen, Denmark; morten.rasmussen@regionh.dk

**Keywords:** colorectal cancer, screening, fecal hemoglobin, cause of death, mortality

## Abstract

**Simple Summary:**

Colorectal cancer (CRC) screening participants with significant traces of hemoglobin in their stool have been reported to have higher mortality and different causes of death (other than CRC) compared to those without. We aimed to investigate these differences among screening participants after 33 years of follow-up. We confirmed that participants with detectable fecal hemoglobin were more likely to die in the study period and to die from different causes, such as cardiovascular and endocrine and hematological diseases, compared to those without detectable fecal hemoglobin. This confirms that fecal hemoglobin may have potential as a marker for diseases not directly related to the colon and rectum and may represent a target for future preventive measures.

**Abstract:**

Fecal hemoglobin (f-Hb) detected by the guaiac fecal occult blood test (gFOBT) may be associated with mortality and cause of death in colorectal cancer (CRC) screening participants. We investigated this association in a randomly selected population of 20,694 participants followed for 33 years. We followed participants from the start of the Hemoccult-II CRC trial in 1985–1986 until December 2018. Data on mortality, cause of death and covariates were retrieved using Danish national registers. We conducted multivariable Cox regressions with time-varying exposure, reporting results as crude and adjusted hazard ratios (aHRs). We identified 1766 patients with at least one positive gFOBT, 946 of whom died in the study period. Most gFOBT-positive participants (93.23%) died of diseases unrelated to CRC and showed higher non-CRC mortality than gFOBT-negative participants (aHR: 1.20, 95% CI 1.10–1.30). Positive gFOBT participants displayed a modest increase in all-cause (aHR: 1.28, 95% CI: 1.18–1.38), CRC (aHR: 4.07, 95% CI: 3.00–5.56), cardiovascular (aHR: 1.22, 95% CI: 1.07–1.39) and endocrine and hematological mortality (aHR: 1.58, 95% CI: 1.19–2.10). In conclusion, we observed an association between positive gFOBT, cause of death and mortality. The presence of f-Hb in the gFOBT might indicate the presence of systemic diseases.

## 1. Introduction

Testing for fecal hemoglobin (f-Hb) in average-risk individuals is an integral part of colorectal cancer (CRC) screening programs [1,2]. Fecal samples are taken to measure f-Hb, often using the nonquantifiable guaiac fecal occult blood test (gFOBT) or the quantifiable fecal immunochemical test (FIT) [3,4,5]. The test result is positive if a sufficient level of hemoglobin is detected, which, for the gFOBT, is approximately 80 µg Hb/g feces [6]. A positive gFOBT result is usually followed by invitation for a diagnostic colonoscopy. F-Hb has been found to be a strong predictor of CRC and a viable initial investigation target in screening programs seeking to reduce CRC mortality [7,8,9,10]. For example, a recent study found that screening participants with a positive gFOBT had higher all-cause mortality rates and were more likely to die from causes other than CRC compared to those with a negative test. This association persisted after adjusting for possible confounding factors, such as age, sex and social deprivation [11]. A Taiwanese research group conducted a similar study among screening participants investigated with the FIT and their risk of cardiovascular disease and found an increase in incidence and mortality rates as FIT levels increased [12].

These results might indicate the possible predictive potential of f-Hb for post-gFOBT/FIT non-CRC survival. However, existing studies lack individual-level adjustment for socioeconomic status, comorbidity and long-term follow-up. Therefore, the aim of this study was to investigate the potential impact of f-Hb on mortality and cause of death by comparing gFOBT-positive and -negative individuals after 33 years of follow-up.

## 2. Materials and Methods

### 2.1. Study Population

We followed participants from the randomized controlled Hemoccult-II (HM-II) trial cohort for more than three decades using the original trial data enriched with data from national registers on health and population [13,14,15]. The HM-II trial enrolled participants in 1985 and 1986 from the Danish region of Funen and then conducted nine rounds of biennial gFOBT-based screening using the Hemoccult-II test, terminating in 2002. Inclusion criteria were age 45–75 and no prior history of CRC or other abdominal surgery. A total of 30,967 people were invited to submit a gFOBT stool sample in the first round of screening and were subsequently invited for colonoscopy if the test was positive. Only participants agreeing to participate in the first screening round were reinvited for subsequent rounds. Written informed consent was obtained from participants before they entered the study in 1985, and the Danish Data Protection Agency approved the study protocol. All individuals with a positive gFOBT were invited for an interview, physical examination and colonoscopy. The study is described in detail elsewhere [14]. In our study, we included all participants from the trial with one or more completed gFOBTs during the trial.

### 2.2. Data Sources

The trial cohort was followed for 33 years. Individual follow-up lasted from the date of inclusion in 1985–1986 and until death, emigration or the 31st of December 2018 (median: 23 years, IQR: 13.8–32.6). Follow-up was conducted using Danish national registers on health and population (Figure 1). Data from the HM-II trial were collected from the Danish National Archives. In addition, we used the Danish Register of Causes of Death (DRCD) to establish cause and time of death [16]. We extracted data from the National Patients Register to identify relevant diseases diagnosed at any Danish public hospital [17]. We used the Danish Education Register [18] and the Income Statistics Register [19] to ascertain socioeconomic status. In the HM-II trial, CRC-related deaths were reviewed using a panel of experts that reviewed all death certificates from 1985 to 2002 where doubt about the actual cause of death was present. These death records were used in the underlying period for CRC deaths. From 2002 onwards, we used the registered cause of death in the DRCD. We used the period from 1980 to 1985 to establish baseline income and conditions. We achieved complete follow-up on all participants for all outcomes but excluded some participants due to missing data on education and income. The raw data are available in the Danish National Archives and the national registers and can be accessed by researchers.

### 2.3. Outcome Measures 

All-cause mortality and the separate causes of death are included as outcomes in this study. All causes of death were recorded using ICD-8 (1985–2001) or ICD-10 (2001–2018). We used the categories of cause of death used by Libby et al. to increase comparability [11]. This included death due to: CRC, non-colorectal cancer, cardiovascular disease, respiratory disease, digestive disease, neuropsychological disease, hematological and endocrine disorders and external causes (Appendix A). External causes included accidents and other non-disease-related causes of death. We used individual-level data to follow all participants from their date of inclusion until death using the DRCD in combination with trial records. We used both the underlying and the primary contributing causes of death registered in the DRCD as our primary outcome measurements. The choice to use both was made to reduce the significance of registration errors in the DRCD [16].

### 2.4. Exposure and Covariate Measurements 

We extracted gFOBT results from all nine rounds of screening and identified all participants with a least one positive test result. Since participants could participate in up to nine rounds of screening and have more than one positive gFOBT during the trial, we found it necessary to account for differences in the accumulated levels of exposure of each participant by introducing gFOBT results as a time-varying exposure. This allowed participants up to nine entries in the dataset with either a negative or a positive test result for each entry. Participants were included over time as a control in the screening rounds in which they tested gFOBT negative and as a case in those where they tested gFOBT positive. By doing so, we handled the individual trajectories of each participant appropriately. 

Each participant underwent a baseline interview and examination from which we extracted age and sex. Age at baseline was divided into three groups “<55”, “55–65” and “>65”. Income and education were both included as socioeconomic status indicators. We divided income into tertiles from lowest to highest based on the five-year average annual income before the inclusion date. The highest completed education was estimated at baseline and divided into “Primary”, “Secondary” and “Higher”. The first category covers elementary school, the second covers high school and vocational educations and the last covers short, medium and long periods of higher education. We included known conditions and diseases suspected of affecting the result of the gFOBT by causing gastrointestinal bleeding up to five years before inclusion (Appendix A). In addition, we adjusted for the effects of comorbidity from the date of inclusion and five years backwards in time using the Charlson Comorbidity Index [20].

### 2.5. Statistics

X^2^-tests were used to compare the gFOBT-positive and -negative participants. We investigated the presence of effect modification on all outcomes for all covariates that were significant in the X^2^-test. Kaplan–Meier curves were used to depict survival. We used Cox proportional hazards regression models considering positive gFOBT as time-varying exposure to estimate the crude and adjusted hazards ratios (HRs and aHRs) and their 95% confidence intervals. We conducted both univariate and multivariate analyses on all outcomes. Log–log plots were used to assess the proportional hazards assumptions, and we excluded outcomes if the assumptions were not met. For our primary analyses, we chose to only use participants with no missing data on any covariates. To investigate the impact of this decision, we also conducted a sensitivity analysis investigating whether the exclusion of all participants with missing values for education impacted our results. All analyses were performed in Stata 16.0 [21].

## 3. Results

### 3.1. Demographics 

During the first round of the HM-II trial, 30,967 people were invited to submit a stool sample. We included all 20,694 (66.8%) participants in the trial who submitted a stool sample in the first round. At the end of our study period, 15,542 (75.1%) had died. They had a mean age at death of 80 (IQR, 74–87) years. Therefore, 1766 (8.5%) gFOBT-positive participants were included in the analysis, representing a total of 1866 positive gFOBTs. One hundred participants had two or more positive gFOBTs during the nine rounds of screening, with a maximum of four. A total of 40.2% of participants were 55–65 years old at inclusion. More females (52.9%) participated than males (46.1%), but more males tested gFOBT positive (54.5%) than females (45.5%). Most of our population had primary education as their highest completed education (42.1%), but many participants had missing registrations on education (27.3%). The Charlson Comorbidity Index showed that 428 participants (2.1%) scored 2 points or higher (signifying extensive comorbidity) at baseline (Table 1). We observed 8566 participants (41.4%) who participated in all nine rounds of screening. A total of 10,070 (48.0%) participants had no missing values and were eligible for analysis. Causes of death are presented in Table 2.

### 3.2. Mortality and Cause of Death 

Kaplan–Meier curves comparing gFOBT-positive to -negative participants showed a difference in mortality rate between the two groups (Figure 2). Participants with a positive gFOBT appear more likely to have died during the course of the study compared to those with a negative gFOBT.

We performed Cox proportional hazards regressions on all participants with no missing values for any covariates. We investigated all nine outcomes and found significant differences between screening participants with a positive gFOBT and those with a negative gFOBT (Figure 3).

Multivariate analyses considering all potential covariates revealed that those testing gFOBT positive were more likely to die in the study period from all causes (aHR: 1.28, 95% CI: 1.18–1.38). We observed an association between CRC and gFOBT results in which those with f-Hb had a higher risk of dying from CRC (aHR: 4.07, 95% CI: 3.00–5.56). Furthermore, the same participants were also more likely to die from all causes, excluding those who died from CRC (aHR: 1.20, 95% CI: 1.10–1.30), and from non-colorectal cancers (aHR: 1.30, 95% CI: 1.12–1.51). Cardiovascular disease as a cause of death was associated with f-Hb (aHR: 1.22, 95% CI: 1.07–1.39). The same was true for endocrine and hematological diseases as the underlying cause of death (aHR: 1.58, 95% CI: 1.19–2.10). We also found an association between respiratory disease as the cause of death and f-Hb (aHR: 1.19, 95% CI: 1.01–1.40). Digestive diseases also appeared to be more common as a cause of death among participants with detectable f-Hb (aHR: 1.50, 95% CI: 1.07–2.10). We did not observe any association between external conditions as a cause of death and the gFOBT result (aHR: 1.09, 95% CI: 0.69–1.74) (Figure 4). The interpretation of the log–log plots on proportional hazards assumptions led to the exclusion of neuropsychological diseases as a cause of death from our analyses.

### 3.3. Sensitivity 

We conducted a sensitivity analysis in which we included those with missing values for the educational level to see if this would affect our results. Although most of the aHRs changed slightly, only the aHR of respiratory disease became statistically insignificant (aHR: 1.13, 95% CI: 0.98–1.30) (Appendix A). 

## 4. Discussion

We investigated the association between f-Hb and mortality in a large, randomized population that, to our knowledge, represents the most extended follow-up in the current literature. Our analyses showed a modest association between detectable f-Hb and mortality that persisted after adjusting for all available confounding factors. We observed an association between f-Hb and death caused by CRC. We also found modest but significant associations between f-Hb and death from other cancers, endocrine and hematological disease, cardiovascular disease, respiratory disease and digestive diseases. Life expectancy did not appear to be significantly shorter in the gFOBT positive.

Recently, the notion that f-Hb may be an indicator for diseases other than CRC was proposed in a study by Chen et al., who found an increased risk of mortality in the population with f-HB [7]. This led to the suggestion that f-Hb may reflect serious non-CRC conditions that affect life expectancy. Findings by Libby et al. supported the results by showing an increased non-CRC-related mortality rate among participants with a positive gFOBT. The authors also reported an association between f-Hb and some causes of death other than CRC. Correcting for medication that could cause gastrointestinal bleeding did not change the conclusions [11]. A recent study from Taiwan by Chien et al. supported the Scottish findings by concluding that f-Hb was associated with cardiovascular mortality [12]. A Korean study supported this by presenting an association between f-Hb and ischemic stroke, myocardial infarction and all-cause mortality [22]. Another Taiwanese study suggests that f-Hb is associated with oral cancer and its precursor lesions [23]. Moreover, f-Hb measured by the FIT has also been associated with cancer in the stomach, small intestine and esophagus [24].

A study by Libby et al. investigated medicine consumption as a proxy measure for disease. They found a strong association between f-Hb and prescription medication for heart disease, hypertension, diabetes and depression [25]. A potential explanation could be that f-Hb may be a surrogate marker for a number of lifestyle-related risk factors, such as a Western lifestyle, and their derived effects on changes to the microbiome. The observed association between f-Hb and increased rate of death may therefore reflect an association between lifestyle and increased mortality. The association between f-Hb and cause of death remained after adjusting for known lifestyle surrogates, income and education, and it is therefore unlikely that lifestyle factors can explain our findings alone. Another potential explanation is that the association between mortality and f-Hb is only partly understood. Libby et al. suggested that the presence of subclinical colonic inflammation could be reflected by f-Hb levels. This state may be a surrogate marker for systemic inflammation and, therefore, also a marker of pathogenesis with an inflammatory component [11]. Supporting this argument is a Taiwanese study that found an association between high f-Hb levels and inflammatory-driven metabolic syndrome [26].

Similar findings were presented by a Japanese study, further strengthening this hypothesis [27]. If later studies confirm that f-Hb may be caused by the subclinical inflammatory state, several personalized treatment measures and initiatives could become available. An example could be a newly suggested approach utilizing the anti-inflammatory potential of the cytotoxic polyacetylenic oxylipins falcarinol and falcarindiol derived from carrots. These substances inhibit the enzyme cyclooxygenase 2 (COX-2), as well as tumor necrosis factor-alpha (TNF-alpha) and interleukin 6 (IL-6), which are members of the proinflammatory cytokine cascade and have been implicated in carcinogenesis [28]. The reported associations in both our and other studies suggest that f-Hb might have clinical potential. Although additional studies are needed, it appears that f-Hb levels can be used as a biomarker for several diseases amenable to preventive measures. In a population-based screening population, participants testing positive for traces of f-Hb with no suspected cause of bleeding detected at a subsequent endoscopy could be a viable group for general follow-up diagnostic initiatives. Future studies exploring the potential of f-Hb in different clinical settings and in combination with disease-specific diagnostic or monitoring modalities are needed. A Chinese study suggested the prognostic use of f-Hb to predict complications and survival after R0 gastrectomy [29]. Another one from Scotland suggested using f-Hb measured by the FIT as a prioritization tool for endoscopic investigations in patients with iron deficiency [30]. These preoperative approaches represent targets for future studies and initiatives that could elaborate on the clinical potential of f-Hb. 

The strengths of this study include a long period of follow-up in a large, randomized population, in addition to the extensive individual-level data retrieved through the Danish registers.

Limitations of our study include the potential misclassification of cases by the DRCD due to the quality of the input data. However, this was addressed by taking contributing causes of death into account for each cause of death. We also have no reason to suspect that potential misclassification is differential. The unquantifiable nature of the gFOBT also represents a weakness, as it does not allow for further stratification into f-Hb levels, which might add valuable insight into the nature of the observed association. Another limitation is the lack of data on prescription medication, obesity and other lifestyle risk factors such as physical activity, smoking and diet.

Although we cannot directly adjust for all of these variables, we believe to have addressed a significant part of the expected effect of between-group differences in lifestyle-related factors by including income and education as proxy variables in our regression analyses. Using prescription medications, such as diabetic and antihypertensive drugs, as proxies for diseases at baseline would have been a reasonable addition to our project. However, since we already adjusted for the effects of comorbidity at baseline using hospital-registered diagnoses, we do not expect that this addition would have a substantial effect on our HRs or the interpretation of our results. Medication causing gastrointestinal bleeding, such as anticoagulants, could influence the risk of a positive gFOBT and would be relevant to adjust for in a multivariate regression model. The only available source for prescription medication when addressing our topic retrospectively is the Danish Medical Statistics Register, which began registering prescriptions in 1994, almost 10 years after our baseline. We were therefore not able to obtain this information. We did, however, adjust for the presence of diseases or conditions that may cause gastrointestinal bleeding at baseline, which should address some of the expected effects on the risk of positive gFOBT. Future prospective studies exploring the associations presented here should address the consumption of relevant drugs as a potential confounding factor. Clinical trials investigating the association between lifestyle factors and f-Hb are needed.

We have a proportion of missing data on educational registrations. We believe that this may result from the lack of registrations in the early years of the register and, in part, the age of the participants at register creation. This was addressed by completing our analysis without these participants, and it was found that it only significantly affected mortality from respiratory disease. The FIT has gradually replaced the gFOBT, which sees little clinical application today. This somewhat limits the direct translation of our results regarding modern CRC screening programs. However, due to the FIT being a recent addition to screening, obtaining a long-term follow-up using a FIT-positive population was not possible, and the gFOBT remains our best alternative.

## 5. Conclusions

We conducted what is, to our knowledge, the most extensive follow-up in a randomized CRC screening population, with more than three decades of follow-up time. We found a modest association between f-Hb measured by the gFOBT and death from several causes other than CRC. Our results indicate that f-Hb may reflect a (patho-) physiological state more prone to morbidity and mortality, reflected by a higher risk of death from other cancers than CRC, as well as endocrine, hematological, cardiovascular and digestive diseases. Limitations include a lack of information on diet, lifestyle, body mass index and prescription medication. Studies to clarify the role of f-Hb and its association with specific diseases are needed.

## Figures and Tables

**Figure 1 cancers-14-00246-f001:**
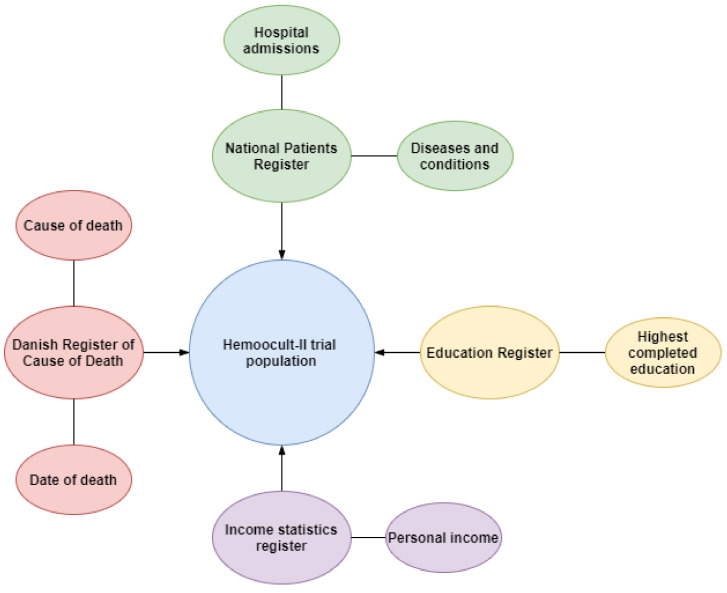
Data sources.

**Figure 2 cancers-14-00246-f002:**
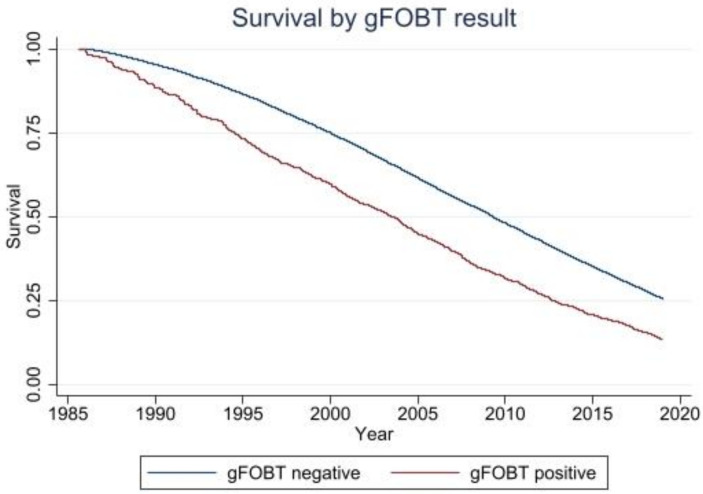
Survival by gFOBT result. Abbreviations: gFOBT, guaiac fecal occult blood test.

**Figure 3 cancers-14-00246-f003:**
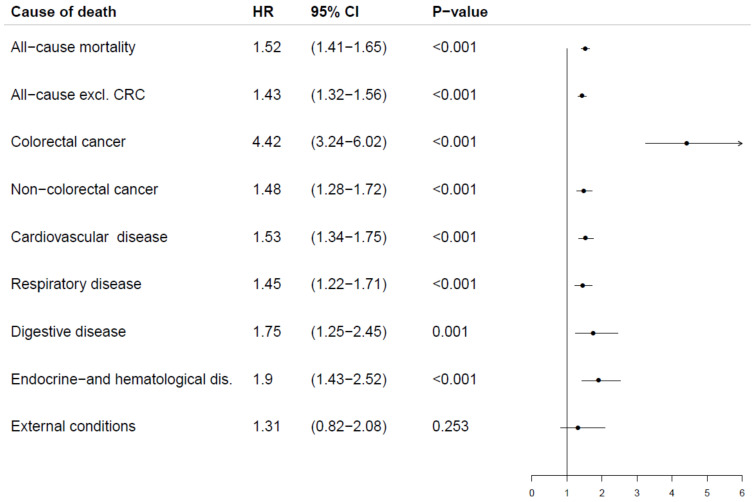
Cause of death and gFOBT result by univariate Cox regressions.

**Figure 4 cancers-14-00246-f004:**
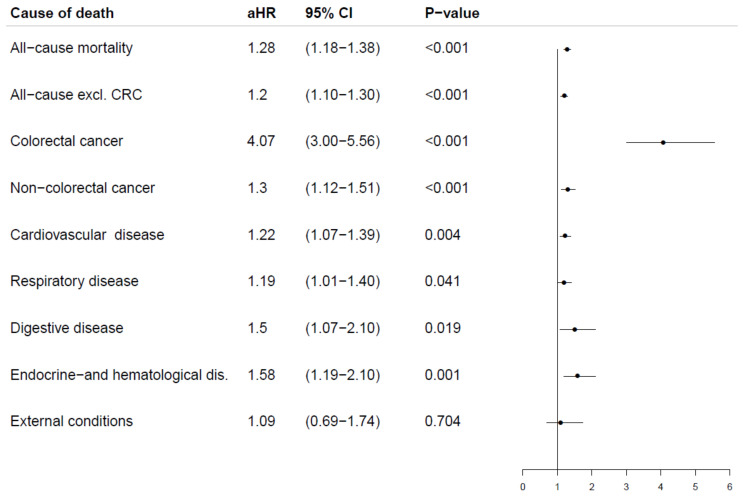
Cause of death and gFOBT result by multivariate Cox regressions. Adjusted for: age, gender, income, education, bleeding at baseline and comorbidity at baseline. Abbreviations: aHR, adjusted hazard ratio; CRC, colorectal cancer.

**Table 1 cancers-14-00246-t001:** Demographics stratified by gFOBT result.

	(+)ve gFOBT (*n* = 1766)	(−)ve gFOBT (*n* = 18,928)	Group Comparison
	N (%)	N (%)	
Gender			
Female	804 (45.53)	10,150 (53.62)	
Male	962 (54.47)	8778 (46.38)	<0.001
Age group at baseline			
<55	573 (32.45)	6471 (34.19)	
55–65	779 (44.11)	7543 (39.85)	
>65	414 (23.44)	4914 (25.96)	0.002
Education			
Primary	749 (42.41)	7972 (42.12)	
Secondary	402 (22.03)	4156 (21.96)	
Higher	157 (8.89)	1618 (8.55)	0.58
Missing data	458 (25.93)	5182 (27.38)	
Income			
1st tertile	554 (31.37)	6340 (33.50)	
2nd tertile	552 (31.26)	6342 (33.51)	
3rd tertile	658 (37.26)	6235 (32.94)	0.002
Missing	<10	<10	
Charlson Comorbidity Index			
0	1708 (96.72)	18,322 (96.80)	
1	24 (1.36)	212 (1.12)	
>2	34 (1.93)	394 (2.08)	0.606
Status			
Alive December 31st 2018	377 (21.35)	4775 (25.23)	
Dead December 31st 2018	1389 (78.65)	14,153 (74.77)	
Age at death (Median, IQR)	81 (75–87)	80 (74–87)	
Conditions suspected of causing bleeding at baseline			
Yes	52 (2.94)	404 (2.13)	
No	1714 (97.06)	18,525 (97.87)	0.027

**Table 2 cancers-14-00246-t002:** Cause of death by gFOBT result for participants with non-missing data.

	(+)ve gFOBT (*n* = 946) (%)	(−)ve gFOBT (*n* = 9122) (%)
All-cause mortality	946 (100.00)	9122 (100.00)
All-cause excl. CRC	882 (93.23)	8813 (96.59)
CRC	64 (6.45)	311 (3.41)
Non-CRC	277 (29.28)	2751 (30.17)
Cardiovascular disease	340 (35.94)	3328 (36.48)
Respiratory disease	236 (24.95)	2160 (23.67)
Digestive disease	57 (6.03)	444 (4.87)
Endocrine and hematological disease	73 (8.35)	595 (6.52)
External conditions	31 (3.28)	335 (3.67)

Abbreviations: gFOBT, guaiac fecal occult blood test; CRC, colorectal cancer.

## Data Availability

All data can be accessed by researchers using a combination of the Danish National Achieves and the Danish registers on health and population listed in Section 2 through Statistics Denmark at https://www.dst.dk/en/ (last accessed on 25 November 2021).

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
