# Peer review of "Cause of Death, Mortality and Occult Blood in Colorectal Cancer Screening"

_cancers, 2022, doi:10.3390/cancers14010246_

Round 1

Reviewer 1 Report

Thank you for amending the manuscript.
Based on the conclusion, how do the authors monitor FOBT positive “healthy individuals”? Along with colonoscopy screening, how often should cardiovascular echography, chest CT scans and endocrine-related tests be performed? It would be better to mention their monitoring method briefly in a clinical setting in order to have general physicians be aware of the importance of FOBT positivity.

Author Response

Thank you for taking the time to review our manuscript.

Reviewer 2 Report

Interesting study. The Authors have tried to modify the paper according to the suggestions

Author Response

Thank you for taking the time to review our manuscript and for you're comments. 

All the best,

Lasse Kaalby
Corresponding author

This manuscript is a resubmission of an earlier submission. The following is a list of the peer review reports and author responses from that submission.

Round 1

Reviewer 1 Report

The authors demonstrated an association between positive gFOBT, cause of death and mortality, leading to the hypothesis that f-Hb might indicate a condition disposing diseases not limited to the colon and rectum. Although this article is interesting, and might be acceptable for publication, there are some points to be addressed. The authors should make the following modifications in order to bolster their claims:

1. It would be better to show in-depth information regarding the degree of obesity and oral medicine including anti-coagulant drugs, antihypertensive drugs, diabetic drugs and so on at the time of positive results.

2. I am not sure about the definition of ‘negative controls’. Are ‘negative controls’ constantly negative for 9 rounds of screening? In addition, the author should present a histogram of the frequency of positivity in gFOBT positive cases. Is there any relationship between the frequency of positivity and the mortality? (For example, the more frequently positive (e.g. more than 3 times) gFOBT is, the higher the mortality is.) In the same vein, did the authors deal cases of positive test just only once as equally as cases of 9 times positive?

3. It would be desirable to use a Charlson Comorbidity Index estimated at the time of positive results, not five years prior to inclusion.

Author Response

Dear reviewer,

Thank you for the valuable questions. Please see our point-by-point reply below.

All the best,

Lasse Kaalby

Response to Reviewer 1 Comments

The authors demonstrated an association between positive gFOBT, cause of death and mortality, leading to the hypothesis that f-Hb might indicate a condition disposing diseases not limited to the colon and rectum. Although this article is interesting, and might be acceptable for publication, there are some points to be addressed. The authors should make the following modifications in order to bolster their claims:

Point 1:

It would be better to show in-depth information regarding the degree of obesity and oral medicine including anti-coagulant drugs, antihypertensive drugs, diabetic drugs and so on at the time of positive results.

Response 1:

Yes, we agree, this type of information would definitely benefit the study greatly. Since this is a register-based study, data on obesity and other related life-style related factors are not obtainable.
Regarding the medicine consumption, we would like to direct your attention to the discussion line 279-280: “We also wished to include prescription medication as a part of the study, but our only source of data is the Danish Medical Statistics Register, which began recording prescriptions in 1994, almost ten years after our baseline”. The lack of prescription medication in our study is the “price” we pay for of obtaining the unique 33-year follow-up.

We have added a line of text to the discussion to elaborate:

“Future prospective studies exploring the associations presented here should address the consumption of relevant drugs as a potential confounding factor.”

Point 2:

I am not sure about the definition of ‘negative controls’. Are ‘negative controls’ constantly negative for 9 rounds of screening? In addition, the author should present a histogram of the frequency of positivity in gFOBT positive cases. Is there any relationship between the frequency of positivity and the mortality? (For example, the more frequently positive (e.g. more than 3 times) gFOBT is, the higher the mortality is.) In the same vein, did the authors deal cases of positive test just only once as equally as cases of 9 times positive?

Response 2:

The negative controls are not constantly negative for all nine rounds of screening. The challenge with multiple rounds of screening is the accumulated exposure. Therefore, to make sure that all participants enter the study properly, they can enter the survival analysis multiple times, one for each time they have a positive gFOBT. This allows for an analysis that strongly reduces the bias that occurs from accumulated exposure. As a result, all participants enter the study with all negative and positive test results.

We are happy to provide a histogram as requested, but would like to point out that only 100 of our patients had more than one positive test, and none had more than two positive tests in total. Instead, we suggest adding this to the results section for clarification.
We have added the line “analysis representing a total of 1,866 positive gFOBTs. 100 participants had two or more positive gFOBTs during the nine rounds of screening with a maximum of four.” to clarify. If a histogram is deemed more relevant, we are happy to provide it.

Point 3:

It would be desirable to use a Charlson Comorbidity Index estimated at the time of positive results, not five years prior to inclusion.

Response 3:

The CCI is already included at the time of positive result. We looked at the patients’ medical history from the date of inclusion and 5 years backwards in time to allow us to properly include pre-existing or chronic conditions. We recognize that clarifications are in order, and we have adjusted the sentence in the methods section:

“In addition, we adjusted for the effects of comorbidity from the date of inclusion and five years backwards in time using a Charlson Comorbidity Index.”

Reviewer 2 Report

The Authors followed up a cohort of individuals who underwent colorectal cancer screening via detection of fecal hemoglobin and reported mortality rate after 33 years of observation. Interestingly a positive test was correlated to a higher mortality from causes different form colorectal cancer. This finding is important since it may provide hints to implement preventive strategies in the future. This is a timely study, with a generally well-crafted structure. I do not have any specific comment.

Author Response

Dear reviewer,

Thank you very much for the feedback. We are happy to hear that our study finds you well.

All the best,

Lasse Kaalby

Reviewer 3 Report

The Authors investigated the differences in long term mortality rates  among screening participants with positive or negative detectable fecal hemoglobin after 33 years of follow‐up.  They found that participants with detectable fecal hemoglobin were more likely to die in the study period and to die from different causes, such as cardio‐vascular and endocrine‐and hematological diseases, than those without detectable fecal hemoglobin. They suggest that fecal hemoglobin may have a potential as a marker for diseases not related  directly to the colon and rectum, and may represented a target for future preventive measures. The Authors followed participants from the start of the Hemoccult‐II CRC trial in 1985 ‐ 1986 until 31 December 2018. Data on mortality, cause of death and covariates were retrieved using Danish national registers. The Authors performed multivariable Cox‐regressions with time varying exposure, reporting  results as crude and adjusted Hazard Ratios (aHR). Among 1,766 patients with at least one 34

positive gFOBT,  946 died in the study period. Most gFOBT positive participants (93.23%)

died of diseases not related to CRC, and showed higher non‐CRC mortality than gFOBT negative  participants (aHR: 1.20, 95%CI 1.10‐1.30). Positive gFOBT participants displayed a modest increase in all‐cause (aHR: 1.28, 95% CI: 1.18‐1.38), CRC (aHR: 4.07, 95% CI: 3.00‐5.56), cardiovascular (aHR: 38 1.22, 95% CI: 1.07‐1.39) and endocrine  and hematological mortality (aHR: 1.58, 95% CI: 1.19‐2.10). 39

In conclusion, the Authors  hypothesize that fHb might indicate a condition disposing diseases not limited to the colon and  rectum.

GENERAL COMMENT

This is a very interesting retrospective analysis including a large number of patients.

The study gives a definitive answer about the fact that positive fecal occult hemoglobin is related with long term increased mortality for not Colorectal cancer causes.

POSITIVE POINTS

The large number of patients and the very long follow up are the most important aspects of the study.

The Authors performed a detailed statistical analysis. Inevitably a retrospective study did not give the possibility to analyze many significant variables, including the presence of possible co morbidities at the time of the first fecal test (positive or negative).

POINTS WHICH I RECOMMEND TO CLARIFY

1-Despite reported differences in mortality rates, the life expectancy in the two groups (positive and negative fecal test) were similar (80 versus 81 years): How do you explain this evidence? The help of a statistician  might clarify this and other statistical aspects of the study.

2-Inevitably a major point is missing: which drugs did the patients take at the time of the fecal tests? We may assume that patients with cardiovascular disease took aspirin and other anti-platelets agents and therefore the possibility exists that their positive fecal test was related to the assumption of anti-platelet agents. Patients with known previous cardiovascular events at the time of the fecal test had higher probability of dying earlier from cardiovascular problems.

The Authors tried to underline this point in the discussion.

3-The question arises also for patients with hematological problems: did they take specific drugs at the time of the positive fecal test?

4-The Authors   hypothesize that fHb might indicate a condition disposing diseases not limited to the colon and  rectum. I would rephrase “THE PRESENCE OF HEMOGLOBIN IN THE FECAL TEST MAY INDICATE THE PRESENCE OF SYSTEMIC DISEASES “.

Liver diseases, assumption of anti-platelet agents, hematological diseases may all give fragility of the colorectal wall and bleeding not related to colorectal cancer .

5-By a clinical point of view, I agree that in presence of positive fecal test, without a clear evidence of colorectal disease, further investigation are warranted to exclude other systemic. However, this test cannot substitute specific tests for the diagnosis of anemia and hematological disorders.

6-I would suggest, if possible, to differentiate mortality rates in patients who had multiple positive fecal tests versus those who had only one positive fecal test.

FINAL COMMENT

This is a very important study which gives almost a conclusive answer about  the statistical significance for associated diseases when hemoglobin  fecal test is positive. Inevitably, the test cannot enter in the clinical setting as substitute of other more specific tests. The clinician should suspect other systemic diseases in patients with a positive fecal tests after having excluded colorectal problems.

Author Response

Dear reviewer,

Thank you for the questions and valuable feedback. Please see our point-by-point reply. 

All the best,

Lasse Kaalby

Response to Reviewer 3 Comments

Comments and Suggestions for Authors

The Authors investigated the differences in long term mortality rates  among screening participants with positive or negative detectable fecal hemoglobin after 33 years of follow‐up.  They found that participants with detectable fecal hemoglobin were more likely to die in the study period and to die from different causes, such as cardio‐vascular and endocrine‐and hematological diseases, than those without detectable fecal hemoglobin. They suggest that fecal hemoglobin may have a potential as a marker for diseases not related  directly to the colon and rectum, and may represented a target for future preventive measures. The Authors followed participants from the start of the Hemoccult‐II CRC trial in 1985 ‐ 1986 until 31 December 2018. Data on mortality, cause of death and covariates were retrieved using Danish national registers. The Authors performed multivariable Cox‐regressions with time varying exposure, reporting  results as crude and adjusted Hazard Ratios (aHR). Among 1,766 patients with at least one 34 positive gFOBT,  946 died in the study period. Most gFOBT positive participants (93.23%) died of diseases not related to CRC, and showed higher non‐CRC mortality than gFOBT negative  participants (aHR: 1.20, 95%CI 1.10‐1.30). Positive gFOBT participants displayed a modest increase in all‐cause (aHR: 1.28, 95% CI: 1.18‐1.38), CRC (aHR: 4.07, 95% CI: 3.00‐5.56), cardiovascular (aHR: 38 1.22, 95% CI: 1.07‐1.39) and endocrine  and hematological mortality (aHR: 1.58, 95% CI: 1.19‐2.10). In conclusion, the Authors  hypothesize that fHb might indicate a condition disposing diseases not limited to the colon and  rectum.

GENERAL COMMENT

This is a very interesting retrospective analysis including a large number of patients.

The study gives a definitive answer about the fact that positive fecal occult hemoglobin is related with long term increased mortality for not Colorectal cancer causes.

POSITIVE POINTS

The large number of patients and the very long follow up are the most important aspects of the study.

The Authors performed a detailed statistical analysis. Inevitably a retrospective study did not give the possibility to analyze many significant variables, including the presence of possible co morbidities at the time of the first fecal test (positive or negative).

POINTS WHICH I RECOMMEND TO CLARIFY

Point 1:

Despite reported differences in mortality rates, the life expectancy in the two groups (positive and negative fecal test) were similar (80 versus 81 years): How do you explain this evidence? The help of a statistician  might clarify this and other statistical aspects of the study.

Response 1:

Thank you, this is a very valid question and the subject needs elaborating.

We have conducted a t-test on the mean differences in the observation time in both groups. This test confirmed a difference between the groups in the mean time-in-study (i.e. from inclusion to death or end of study) which was 22,2 years for the gFOBT negative and 17,9 years for the gFOBT participants. This confirms that although the life expectancy were higher among the gFOBT positives, they are more likely to die in the study period.

We believe that the differences in life expectancy may be explained by the differences in the frequency of participants that have died in the two groups. More gFOBT negative than positive participants are still alive. In the coming years, more deaths may equalize the differences between the groups. We have added a sentence in the discussion to clarify:

“…life expectancy did not appear to be significantly shorter in the gFOBT positive.”

Point 2:

Inevitably a major point is missing: which drugs did the patients take at the time of the fecal tests? We may assume that patients with cardiovascular disease took aspirin and other anti-platelets agents and therefore the possibility exists that their positive fecal test was related to the assumption of anti-platelet agents. Patients with known previous cardiovascular events at the time of the fecal test had higher probability of dying earlier from cardiovascular problems. The Authors tried to underline this point in the discussion.

Response 2:

This is, unfortunately, a limitation of this study. It would have been extremely interesting to include this, but the data does sadly not exist as stated in the discussion.

We would, however, like to point out that the only other directly comparable study by Libby et al. did include prescription drugs, and still observed an association between f-Hb and cause of death. The inclusion of prescription medication would probably have a slight effect on our results, but we do not suspect it to change the overall conclusion. By introducing comorbidity as a covariate, we also believe to have adjusted for some of the effect originating from prescription medication.

We have added a line of text to elaborate on the importance of this in the discussion:

“Future prospective studies exploring the associations presented here should address the consumption of relevant drugs as a potential confounding factor.”

Point 3:

The question arises also for patients with hematological problems: did they take specific drugs at the time of the positive fecal test?

Response 3:

This is a valid point. As previously mentioned, we do not suspect that the inclusion of prescription medication as a covariate would change the conclusions of our study.

Point 4:

The Authors   hypothesize that fHb might indicate a condition disposing diseases not limited to the colon and  rectum. I would rephrase “THE PRESENCE OF HEMOGLOBIN IN THE FECAL TEST MAY INDICATE THE PRESENCE OF SYSTEMIC DISEASES “. Liver diseases, assumption of anti-platelet agents, hematological diseases may all give fragility of the colorectal wall and bleeding not related to colorectal cancer.

Response 4:

We have rephrased the finishing sentence of the abstract to meet the suggestion:

We hypothesize that the presence of f-Hb in the gFOBT may indicate the presence of systemic diseases.”

Point 5:

By a clinical point of view, I agree that in presence of positive fecal test, without a clear evidence of colorectal disease, further investigation are warranted to exclude other systemic. However, this test cannot substitute specific tests for the diagnosis of anemia and hematological disorders. 

Response 5:

We agree. Our point with the warranted investigations was directed more towards a general need for a visit to e.g. a GP that should then decide which follow-up tests are relevant. Therefore, we do not suggest substitution of any current modalities. We have added a short elaboration in line 262:

“…could be a viable group for general follow-up diagnostic initiatives.”

Point 6:

I would suggest, if possible, to differentiate mortality rates in patients who had multiple positive faecal tests versus those who had only one positive faecal test.

Response 6:

The use of time-varying exposures allows each patient to contribute to the mortality rate with different levels of exposure (i.e. positive gFOBTs). Therefore, our analysis already account for the number of positive gFOBTs when mortality rates are calculated. We implemented this approach to reduce the effect of exposure levels between patient groups.
We have explored the options on stratification by number of positive test, and found that since the number of participants with more than one positive test was low, we did not have sufficient statistical power. We explored the life expectancy after first positive gFOBT and found that having only one positive test did not improve the mortality rate compared to those with more than one.

As a result, we believe that a stratification by gFOBT result as suggested would not be beneficial for the interpretation of our results, as the intended effect is already accounted for. To elaborate, we have added the following sentence in the result section:

… representing a total of 1,866 positive gFOBTs. 100 participants had two or more positive gFOBTs during the nine rounds of screening with a maximum of four.”

FINAL COMMENT

This is a very important study which gives almost a conclusive answer about  the statistical significance for associated diseases when hemoglobin  fecal test is positive. Inevitably, the test cannot enter in the clinical setting as substitute of other more specific tests. The clinician should suspect other systemic diseases in patients with a positive fecal tests after having excluded colorectal problems.

Round 2

Reviewer 1 Report

In my opinion, in order to draw an accurate conclusion, it is essential to add in-depth information regarding the degree of obesity and oral medicine including anti-coagulant drugs, antihypertensive drugs, diabetic drugs and so on at the time of positive results.